# Towards Optimum Mandibular Reconstruction for Dental Occlusal Rehabilitation: From Preoperative Virtual Surgery to Autogenous Particulate Cancellous Bone and Marrow Graft with Custom-Made Titanium Mesh—A Retrospective Study

**DOI:** 10.3390/jcm12031122

**Published:** 2023-01-31

**Authors:** Kei Onodera, Ikuya Miyamoto, Isao Hoshi, Shinsuke Kawamata, Noriaki Takahashi, Nobuko Shimazaki, Hisatomo Kondo, Hiroyuki Yamada

**Affiliations:** 1Division of Oral and Maxillofacial Surgery, Department of Oral and Maxillofacial Reconstructive Surgery, Faculty of Dental Medicine, Iwate Medical University, 19-1 Uchimaru, Morioka 020-8505, Iwate, Japan; 2Division of Oral and Maxillofacial Radiology, Department of Oral and Maxillofacial Reconstructive Surgery, Faculty of Dental Medicine, Iwate Medical University, 19-1 Uchimaru, Morioka 020-8505, Iwate, Japan; 3Department of Prosthodontics and Oral Implantology, School of Dentistry, Iwate Medical University, 19-1 Uchimaru, Morioka 020-8505, Iwate, Japan

**Keywords:** mandibular reconstruction, virtual surgery, custom-made titanium mesh, autogenous particulate cancellous bone and marrow graft, mandibular condyle displacement

## Abstract

The purpose of this retrospective study was to evaluate computer-assisted virtual surgery and the outcomes of mandibular reconstruction using an autogenous particulate cancellous bone and marrow (PCBM) graft combined with a custom-made titanium mesh (TiMesh) using a three-dimensional (3D) printing model. Eighteen consecutive patients were included, and preoperative virtual simulation surgery was performed using digital data. Segmental bone defects showed deviation of the mandible due to displacement of the condyle and segments, unnatural length of the mandibular body, or poorer intermaxillary relationship compared to the marginal bone defect caused by previous operations. These mandibular disharmonies could be simulated, and virtual surgery was performed on a computer with adjustment of displaced mandibular segments, length of the mandibular body, and dental arch with digital bone augmentation. TiMesh was manually pre-bent using a 3D printing model, and PCBM from the iliac crest was grafted with TiMesh. The short-term clinical results were good; reconstruction of the alveolar crest was prosthetically desirable; and minor complications were observed. In conclusion, virtual reconstruction is crucial for treating complex deviated mandibles. Accurate condylar and dental arch positions with an optimum mandibular length are important for prosthetically satisfactory mandibular reconstruction.

## 1. Introduction

Mandibular reconstruction is performed when there is a functional or aesthetic loss of structures in the oral cavity [1]. There are numerous treatment modalities for mandibular defects, including autogenous bone graft or microvascular bone graft, and each method has its own strengths and weaknesses [2,3,4,5]. Dental implants are frequently used for dental occlusal rehabilitation in reconstructed bones [6,7,8].

After a large mandibulectomy, microvascular bone grafts are preferred for reconstruction [9,10,11]. The drawback of microvascular bone grafts is that the technique involves a long operating time, significant morbidity, and difficulties in reconstructing the complex morphology of the jawbone, especially the alveolar ridge, for dental rehabilitation [12,13]. Ideally, the bone defect would be reconstructed concurrently with the primary disease operation; however, simultaneous surgery is occasionally difficult owing to various reasons involving the patient, operator, and environment. Complications, such as a surgical plate fracture may occur after reconstruction, requiring another operation [14]. An insufficient reconstruction might result in esthetic or functional problems, such as deviation of the mandible, deficiency of the inter-maxillary relationship, or insufficient bone height or width for prosthetic treatment [15]. Furthermore, several studies have reported a low rate of dental implant placement after vascularized bone grafts [16,17,18]. One reason for the difficulty of implant treatment is the poor condition of the alveolar ridge. For optimal prosthetic rehabilitation, additional reconstruction of the alveolar ridge is required.

One of the alternative reconstruction methods is to use a particulate cancellous bone and marrow (PCBM) graft combined with custom-made titanium mesh (TiMesh). PCBM-TiMesh was first described by Boyne [19]. Several studies have reported successful results with this procedure [20,21,22,23,24]. The main advantage of this method is that the reconstructed mandible can be given any morphology with TiMesh bending; thus, it is possible to reconstruct the form as required. Skilful techniques are needed to fabricate the jawbone morphology with TiMesh during surgery, and the exact shape or position of the segmented mandible might be difficult.

To overcome these drawbacks, a computer-assisted virtual surgical simulation and a three-dimensional (3D) printed model with digital data were applied preoperatively. Computer technology has meliorated in recent years, but there are only a few reports on computer-assisted mandibular reconstruction using PCBM and TiMesh [25,26,27,28,29].

The purpose of this retrospective study was to evaluate the simulation using computer-assisted virtual reconstruction surgery and the outcomes of mandibular reconstruction with an autogenous PCBM graft with custom-made TiMesh using a 3D-printed model.

## 2. Materials and Methods

This single-centre, non-interventional, retrospective study was conducted at the Division of Oral and Maxillofacial Surgery, Iwate Medical University, between 1 January 2017, and 31 December 2021, following the principles stated in the Declaration of Helsinki (1964) and its later amendments. Iwate Medical University is an urban university hospital that serves approximately 1,200,000 people in the Iwate prefecture, and approximately 3000 new patients and 15,000–16,000 returning patients visit the division of Oral and Maxillofacial Surgery annually. The study design was approved by the Ethics Committee of the Faculty of Dental Medicine at Iwate Medical University (01305). The STROBE guidelines were followed in this investigation [30]. 

### 2.1. Participants

Eighteen consecutive patients underwent mandibular reconstruction using a PCBM graft with custom-made TiMesh by a single surgical team using computer simulation and a 3D printing model in our department. Patients comprised 13 men and 5 women, with an average age of 60.9 years (range 17–81 years). All patients received written information regarding the surgery and provided written informed consent. All procedures were performed under general anaesthesia.

### 2.2. Inclusion Criteria and Exclusion Criteria

No patients underwent radiation therapy. Mandibular defects were caused by the resection of benign tumours in 8 patients and malignant tumours in 10 patients. The eight benign tumours included five patients with ameloblastomas, one with an odontogenic myxoma, one with an osteoblastoma, and one with an ossifying fibroma. The 10 patients with malignant tumours included 7 lower gingival squamous cell carcinomas, 2 squamous cell carcinomas of the floor of the mouth, and 1 mucoepidermoid carcinoma of the floor of the mouth. One patient (Case 8) underwent reconstruction simultaneously with the abrasive surgery. The primary reconstruction of the segmental defect cases was performed using a rectus abdominis musculocutaneous flap with a surgical plate (case 3), one vascularized fibular flap (case 9), one vascularized scapular flap (case 15), and eight surgical plates. The cases with marginal bone resection were reconstructed using one infrahyoid myocutaneous flap (case 4), while the other five cases did not undergo reconstruction. 

### 2.3. Mandibular Defect Classification

The mandibular defects were classified as marginal bone defects (6 cases, 33.3%) and segmental bone defects (12 cases, 66.7%). In addition to marginal bone resection, Brown’s classification of segmental bone defects was also applied. Five were class I, three were class II, and four were class III [31]. None of the patients underwent mandibulectomy with the condyle. However, there was one case of condylar dislocation (Case 15). The patient characteristics are presented in Table 1.

### 2.4. Mandibular Data Acquisition and the Making of a Virtual Neo-Mandible for Optimum Dental Occlusion

All patients underwent cone-beam-computed tomography (CBCT) (3D Accuitomo F17, MORITA, Kyoto, Japan) preoperatively, including the mandibular condyle and maxilla. The first step consisted of obtaining Digital Imaging and Communications in Medicine (DICOM) files from CT data. The 2D CT images were reconstructed into 3D CT images using the Volume Extractor^®^ software (Volume Extractor 3.0, i-Plants Systems, Iwate, Japan) [32,33] (Figure 1A). Using this software, images were processed to remove artifacts and unwanted structures. After this process, the software was transformed into a digital 3D object file (Standard Tesselation Language, or STL). The STL file was imported into Geomagic Freeform Software (3D Systems, Rock Hill, SC, USA), where it became “digital clay”. A Geomagic Touch haptic device (3D Systems) was used to modify the peripheral artifacts and subtract several unwanted objects. With this preparation, a computer simulation was performed by virtual reconstructive surgeons (K.O., I.H., and S.K.) (Figure 1B). Additionally, virtual simulations of occlusal rehabilitation were performed to establish occlusion on the computer. After virtual bone augmentation, dental implant placement simulation or virtual conventional prosthetic treatments were continued with consideration of the opposing dentition. In some cases, after the computer simulation, these data were used to confirm the occlusal relationship on a 3D-printed model or a conventional individual dental cast model. The cases included a variety of prosthetic conditions with conventional dentures, dental implant prostheses or no prostheses. In some cases, the patient’s requirements changed as the mandibular and alveolar reconstruction progressed with increasing patient motivation for occlusal reconstruction. As the present research aimed to evaluate the PCBM graft with TiMesh using virtual reconstruction surgery, the following research will clarify the establishment of occlusal reconstruction. Implant installation simulations may also be performed if dental implant treatment is a prerequisite. Digital data were imported into the implant planning software (CoDiagnostiX, Straumann, Basel, Switzerland) by prosthodontists (H.K., N.S.) (Figure 1C).

### 2.5. 3D Printing Model Fabrication and Titanium Mesh Pre-Bending

Based on the mandibular data after the virtual surgical simulation, a neo-mandibular 3D model and guide templates, including a mandibulectomy and osteotomy guide template, were produced by K.O. with the use of 3D printing technology (Z printer^®^ 450, Z Corporation, Burlington, Massachusetts, USA) (Figure 1D). Since the plaster can be coloured in full, the healthy bone was distinguished from the lesion and the part to be reconstructed with the grafted bone (Figure 1D). The plaster-based 3D model was coated with an acrylic resin hardener on all surfaces to increase its strength (Z-BOND^TM^90, Z Corporation, Burlington, MA, USA). In the same way, several of the cases were modelled with acrylic resin (Figure 1E). After this preparation, the titanium mesh was pre-bent on the reconstructive neo-mandibular model (0.6 mm thickness, Leibinger Dynamic Mesh, Stryker Japan, Osaka, Japan) (Figure 1F). When there was concern about lower mechanical strength with a single mesh, two mesh layers were used. The joints of the processed mesh were laser-welded. The process from CT imaging to titanium mesh fabrication is shown in Figure 1A–F.

### 2.6. Positioning Gauge and Cutting Guide

In the complicated simulation case (Figure 2A), following the pre-bending of TiMesh, the devices for the cutting guide (Figure 2B,C), positioning gauge (Figure 2D,E), and occlusal splint (Figure 2F) with the anatomical landmarks were prepared individually to increase the accuracy of surgery (Figure 2A–F).

### 2.7. The Detailed Virtual Reconstruction Technique

Our simulation procedure focused on three key points: the condylar position, mandibular length, and dental arch (Figure 3). The simulations were classified into three levels, depending on the degree of mandibular defect or deviation.

#### 2.7.1. Level 1

Level 1 cases are simple bony defects with no condylar deviation or abnormal occlusal relationships. This category is usually observed during simultaneous reconstruction with marginal or segmental mandibular resection. Anatomical features of the normal mandible and their spatial relationships were carefully delineated and restored on an individual basis. The virtual mandible was built using the mirroring tools of the software to interpolate the defect span. The mirrored images were superimposed on the original images using different colours, and the defects were covered by the mirrored images (Figure 4). The shape and position of the mirrored images were adjusted digitally until the mirrored portions of the defects corresponded well with the residual mandibular structure, condyle, and dental arch. 

#### 2.7.2. Level 2

Level 2 cases include mandibular and condylar deviations with bone defects. The patient’s original jawbone information is available with CT data. Therefore, simulation was used to adjust the mandible and condylar position on the computer and then perform bone augmentation digitally. In some cases, occlusal relationships were simulated by capturing information from dental cast models, 3D dental scanners, or dental implant simulations. Using software, the deviated condyle was virtually adjusted (Figure 5). Additionally, the dental arch was virtually adjusted. If the condyle and dental arch did not overlap well, the length and position of the dental arch may be different. Therefore, the virtual augmented mandible was digitally determined. In cases of mandibular length shortage and/or malposition of the mandibular body, the mandible is needed to separate not only the desirable morphology of the mandible but also the physiological condylar position and an accurate intermaxillary relationship. The previously reconstructed bone was virtually cut, and ideal mandibular shapes were reconstructed by adjusting the positions of the condyle, mandibular body, and dental arch (Figure 6). In such cases, cutting guides, positioning gauze, and occlusal splints are useful for accurate surgical positioning.

#### 2.7.3. Level 3

Level 3 was the most difficult. The mandibular and condylar positions deviated, and significant bone defects were observed. Furthermore, these are cases in which preoperative mandibular CT information does not exist because the tumour has destroyed the jaw, or the size of the jaw has changed owing to growth. In these cases, the remaining mandible and condyle are aligned. Subsequently, a virtual jaw (neo-mandible) was created digitally on the computer. The neo-mandible was superimposed to simulate a reconstructed mandible. If necessary, information regarding the maxillary and mandibular dental arches was incorporated into the simulated jawbone. In such cases, mirroring might not be indicated in cases where preoperative CT or dentition information is not available, or in cases of large anterior defects, including bilateral bone defects (Figure 7). Because mirroring was impossible, a digital operation was performed by adapting a dental arch from the individual intraoral scanner, a dental cast model, or a virtual imaginary dental arch to a virtual mandible. This means that virtual reconstruction was performed by fitting a digital dental arch and mandible (Figure 8). After these processes, virtual bone augmentation was performed to allow for prosthetic treatment. Thus, a virtually ideal mandible can be achieved. 

### 2.8. Clinical Procedures 

The PCBM was obtained from the posterior iliac crest, as described by Iino et al. [34]. In all cases, the previously resected extraoral approach was performed using the same incision. In cases requiring mandibular osteotomies, a cutting guide device was used for an accurate osteotomy procedure using piezosurgery. The prefabricated TiMesh was then adapted to the remaining bone using a prefabricated positional guide for accurate fixation. The mesh was fixed to each side of the host bone using 5–8 monocortical titanium screws (diameter: 2 mm; lengths: 5 mm, 6 mm, and 8 mm). The surrounding soft tissues, including muscle, fatty, and fibrous tissues, were then tightly sutured to the lateral, medial, and inferior surfaces of the tray to eliminate any dead space. The PCBM was loaded into the mesh and condensed to increase graft density. After adequate subcutaneous relaxation incisions were made, the overlying skin was sutured with minimal tension. Intermaxillary fixation using an elastic band was maintained for 7–10 days in patients with dentition. An occlusal soft splint was placed in the opposing dental arch to avoid occlusal trauma to the oral membrane of the PCBM. 

### 2.9. Evaluation of Mandible 

The pre- and postoperative mandibles were evaluated digitally using CT data. The length between the mandibular plane and condylar axial angle was measured. The condylar axial angle was defined as the angle between the condylar axial plane (angle between the condylar axis, a line from LCo-MCo) and the midsagittal reference line (line from the Nasion to Basion). LCo is the maximum convex curvature on the lateral aspect of the condyle, and MCo is the maximum convex curvature on the mesial aspect of the condyle [35]. The difference between the sum of the preoperative condylar axial angles on both sides and the sum of the postoperative condylar axial angles was recorded as the change.

### 2.10. Statistical Analysis

Statistical analysis was performed using the Mann–Whitney U test using computer software (EZR) (Saitama Medical Centre, Jichi Medical University, Saitama, Japan) and a graphical user interface for R (The R Foundation for Statistical Computing, Vienna, Austria) [36]. A *p* < 0.05 was considered statistically significant.

## 3. Results

### 3.1. Classification of Simulation

There were seven level 1 cases, nine level 2 cases, and two level 3 simulations (Table 2). 

### 3.2. Operation

The titanium mesh was accurately matched in the mandible between computer simulation and actual operation, and the operation was performed with high accuracy and safety as shown in Figure 9, Figure 10, Figure 11, Figure 12 and Figure 13 corresponding to Figure 4, Figure 5, Figure 6, Figure 7 and Figure 8. The pre-bent titanium mesh was placed without any adjustment, and PCBM placement was performed smoothly. The amount of bone required was predicted using software. However, the amount of iliac bone required was greater than that required in the simulation. Table 2 presents the results of this study.

### 3.3. Evaluation of Reconstructed Mandible

Table 2 presents these data. The mean augmented length of the mandible was 3.21 ± 4.94 (SD) mm. For the marginal bone augmentation, the mean length was −0.15 ± 0.37 mm, whereas for the segmental defects, the mean length was 4.89 ± 5.34 mm (Figure 14). There were statistically significant differences between marginal and segmental bone defects (Mann–Whitney U test, *P* < 0.05).

The mean overall angle of the deviated and repositioned mandible was 4.39 ± 6.70°. For the marginal bone augmentation group, the mean degree of the mandible was 0.33 ± 0.52°, whereas, for segmental defects, the mean was 6.41 ± 7.46°. (Figure 15). There were statistically significant differences between marginal bone defects and segmental defects (Mann–Whitney U test, *P* < 0.05).

### 3.4. Postoperative Complication

After surgery, there were no problematic complications except in two cases. There were no cases of postoperative fatal infection, but there were two cases of infection of grafted bone several months after the operation (Figure 16A). Two cases showed prolonged titanium mesh exposure after grafting, and pus discharge was observed (cases 9 and 13). However, titanium mesh exposure did not affect the outcomes. In these cases, the TiMesh was removed and PCBM was grafted again from the anterior iliac crest into the area of resorption. Mesh exposure was occasionally observed (Figure 16B). In these cases, the exposed part of the mesh was excised under local anaesthesia. Other issues observed were narrowing of the oral vestibule (Figure 16C) or oral floor and the lack of attached gingiva around the alveolar crest, and small bone gaps between the remaining alveolar bone and the grafted bone (Figure 16D). 

There were no functional problems in the short observational period, as there were no patients with trismus or symptoms of TMJ dysfunction. The bone graft worked well, and the optimum bone augmentation for occlusal rehabilitation was achieved.

## 4. Discussion

In the present study, mandibular reconstructions were successfully performed, providing recovery of the mandibular complex deviation and an optimum alveolar bone condition for prosthetic rehabilitation. With the virtual reconstruction concept, the methodology was particularly useful for complicated deviated mandibles caused by previous operations. Furthermore, prefabricated, custom-made TiMesh improved reconstruction accuracy and reduced the morbidity associated with surgical interventions.

One of the important points of the present simulation is that numerous cases showed displacement of the condylar position due to the segmentation of the mandible. Due to muscle contraction, the position of the condyle would have moved. Conventional treatment methods, such as locking occlusion, pre-bending titanium plates, and surgical navigation, can show issues such as low accuracy, being time-consuming, and being difficult to master [37].

Clinically, precise confirmation of the dislocated condylar position is practically impossible during surgery. Furthermore, in virtual surgery, the mirroring method might be problematic in cases of bilateral bone defects because there is no mandibular information to copy. This problem is the same situation as that in the vascularized bone graft, a typical case in the mentum area of the mandible shown in Figure 7 and Figure 12. With our PCBM and TiMesh procedures, it was relatively easy to reconstruct the mandible as designed [22,23,24]. If the CT data or plaster model information of the dentition is available, it can be used as a digital reference for virtual reconstruction. Although no original mandibular information was available, a virtual mandible with a virtual dental arch could be used. To the best of our knowledge, this is the first report on the adaptation of a virtual mandible to PCBM with TiMesh. In short, patients might be able to decide on the design of their mandible or optimal facial appearance by themselves preoperatively.

Another advantage of virtual surgery is that the psychological stress of the surgeon is reduced by visualizing a part of the procedure that relies on the surgeon’s experience and intuition. These results suggest that the simulation of hard tissue using a 3D model is highly accurate and useful [38]. These computer-assisted diagnoses could reveal the facts preoperatively about mandibular disharmony and would have precise surgical accuracy. A virtual reconstruction chart is shown in Figure 17. These are the preliminary results of our study, and there are various conditions corresponding to patient-specific bone defects.

With anatomical consideration, to reproduce the functional mandible accurately, three elements are important, as described above: the position of the mandibular condyle, the size of the mandibular body, and the dental arch with the optimum intermaxillary relationship: dental occlusion (Figure 3).

If these points are insufficient, precise reconstruction is difficult. In particular, mandibulectomy with the condyle was a difficult procedure that was not present in this study. Daves et al. reported the difficulty of the mirroring technique in cases of pathology involving the condyle or coronoid processes [39]. In addition, reconstruction of the mandibular condyle is difficult with microvascular bone grafts [40,41]. In our case, extreme resection of the mandibular ramus showed good results since the condyle was removed from the mandibular fossa during surgery (Case 8, Figure 4 and Figure 9). Another case (Case 15, Figure 8 and Figure 13) showed displacement of the condyle due to previous surgery. In this case, the above three points were insufficient, and the reconstruction was planned with lengthening of the mandibular body and intermaxillary fixation without displaced condylar repositioning due to the refusal of TMJ surgery. The clinical results were satisfactory, with little functional impairment and a high degree of patient satisfaction.

Considering the above three elements, condyle reconstruction might not be necessary in cases where dental occlusion and mandibular length are acceptable. Although some deviation of the mandible might be observed in maximum mouth opening, there was no functional problem with normal masticatory function or daily life, which is in agreement with Chao et al. [42]. In other words, functional problems may occur when two or more points are compromised.

However, there were many cases of condylar displacement, and none of the patients complained of postoperative TMJ dysfunction or pain. Sudden changes in the position of the mandibular condyle could lead to TMJ dysfunction; rather, it is logical that the displaced condyle returns to its original physiological position. Therefore, to minimize functional problems, it is important to reconstruct the mandible with an ideal length and optimum intermaxillary relationship with an alveolar ridge for the establishment of dental occlusion. In the present study, cases with large mandibular reconstructions or fractured surgical plates tended to have deviated mandibles, whereas cases with relatively small reconstructions had a less deviated mandible.

Taken together, it is desirable to preserve the mandibular condyle as much as possible; however, in some cases, it may not be necessary to reconstruct the mandibular condyle if the appropriate mandibular length and dental occlusion have been established. Our cohort was small and limited, and future studies are required to confirm this hypothesis. Reconstruction of the mandibular condyle is currently an unresolved issue.

From a surgical point of view, the PCBM and TiMesh procedures cannot be performed simultaneously in major surgeries involving extensive soft tissue loss. Free flap reconstruction is an important procedure for the reconstruction of facial appearance, mandibular body, and soft tissue defects. However, it is often difficult to simultaneously reconstruct the ideal alveolar ridge with microvascular bone reconstruction alone. PCBM and TiMesh might be useful in compensating for the weakness of microvascular bone grafts, as the operation technique is relatively simple to use to improve the quality of life of patients. PCBM with TiMesh is not in competition with a microvascular bone graft but could be a complementary option. As the PCBM was obtained from the posterior iliac crest, the method required conversion of the abdominal position to the supine position; therefore, the operation time tended to be longer. The amount of extracted bone was measured using a syringe. Consequently, the measured volume was approximately twice the predicted volume. However, the planned bone volume did not match the actual bone volume. At present, it is unclear why these differences have occurred.

At the point of postoperative complication, there were no fatal complications in the present study. Even if the mesh was removed because of infection, partial bone necrosis or resorption occurred, and total bone resorption was not observed. Extraoral incisions showed less infection than intraoral incisions, which had a thin oral membrane, and wound dehiscence and/or exposure to TiMesh were sometimes found [43]. After primary healing, mesh exposure did not directly result in significant bone resorption, and the mesh seemed to tolerate infection.

There were soft tissue problems, including a lack of oral membrane, narrowing of the oral vestibule, and/or oral floor. In the case of a composite free flap, additional thinning or correction of the overlying skin paddle is sometimes necessary [44]. The presence of subcutaneous tissue and the absence of keratinized gingiva could affect implant survival and peri-implant health [13]. The other problem with soft tissue or oral membranes is the narrowing of the oral vestibule and/or floor of the mouth. We performed total mandibuloplasty with a palatal mucosal graft [45]. However, due to the limited amount of mucosa, ideal soft tissue surgery, including peri-implant tissue, is a challenge for the future. The soft-tissue shortage is a decisive factor for QOL after mandibular reconstruction surgery [46]. An alternative problem is a gap between the gingiva and the oral membrane of the grafted bone. It was difficult to equalize the height of the remaining alveolar bone with that of the grafted bone. Elevation of the thin oral mucosa during bone grafting might lead to damage to the oral mucosa. This could lead to traffic between the grafted bone and the oral cavity, which increases the probability of infection of the grafted bone. Therefore, the elevation of the oral mucosa should not be overdone. Thus, the gap between the alveolar bone and the grafted bone seems to be an issue that should be addressed in pre-prosthetic surgery. This study was an attempt to achieve ideal alveolar bone reconstruction for better occlusal reconstruction. We are currently continuing to reconstruct the occlusal rehabilitation with prosthetic treatment using implants on the PCBM-TiMesh reconstructed alveolar bone using digital technology. Our next report is planned on changes in the grafted bone over time, pre-prosthetic surgery of the oral vestibule and floor of the mouth, procedures with the oral mucosa, implant placements, and subsequent prosthetic treatments.

In this study, the bending of the titanium mesh was performed manually. In future research, we plan to create a titanium mesh with digital technology by applying computer-aided design/computer-aided manufacturing (CAD/CAM) technology [47]. Using digital technology, a custom-made titanium mesh with sufficient strength corresponding to the bone defect of an individual patient can be manufactured preoperatively. Using digital technology for a large part of the reconstructive procedure, a future reconstructive operation might be expected to be accurate, minimally invasive, and safe from diagnosis to treatment.

The retrospective study design, small number of patients, and non-randomized patient selection were the major limitations of this study. To establish this operational concept more broadly, large-scale clinical studies using computer-assisted virtual simulation and computer-assisted prefabricated titanium meshes should be conducted.

## 5. Conclusions

Simulation using computer-assisted virtual reconstruction is a crucial technique for repositioning the deviated mandible to the correct position. The following factors contribute to optimal mandibular reconstruction: a precise position of the condyle, a natural length of the mandible, and sufficient alveolar bone for stable dental occlusion.

## Figures and Tables

**Figure 1 jcm-12-01122-f001:**
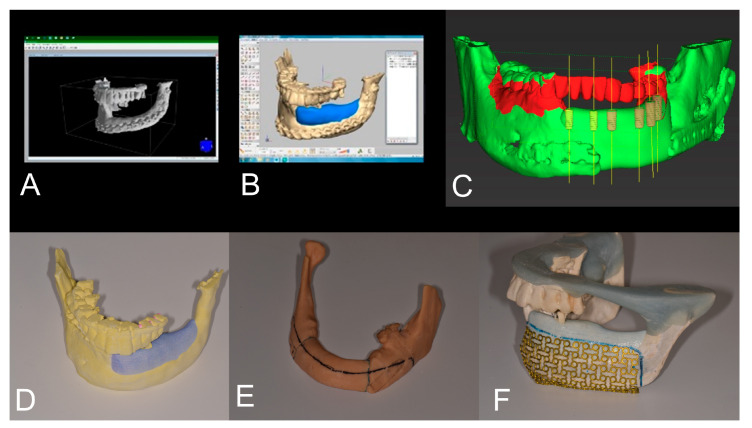
The procedures of virtual mandibular reconstruction with CT data. The first step consisted of obtaining the DICOM files from CT data. The 2D CT images were reconstructed into 3D CT images with the use of Volume Extractor^®^ software (**A**). After this process, the software was used to transform them into a digital 3D object file (Standard Tessellation Language (STL)). The STL file was imported into Geomagic Freeform Software (**B**). In some cases, dental implant simulation was also performed simultaneously (**C**). After the virtual simulation, a mandibular 3D model was produced. The photograph shows a plaster-based 3D model (**D**) and an acrylic resin-based 3D model (**E**). Then, the titanium mesh was pre-bended on the model (**F**).

**Figure 2 jcm-12-01122-f002:**
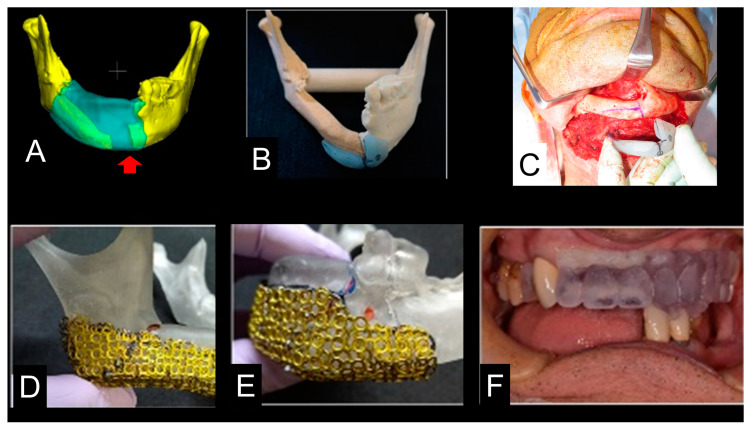
Virtual mandibular reconstruction (Case 11) suggested the cutting points of the mandible (arrow) (**A**). The 3D model was made and the cutting device showed the cutting points (**B**). Clinical application of cutting device (**C**). Titanium mesh was made using double-layer titanium mesh via titanium laser welding with anatomical landmarks (**D**,**E**). Additionally, the occlusal splint was made for accurate occlusal stabilization (**F**).

**Figure 3 jcm-12-01122-f003:**
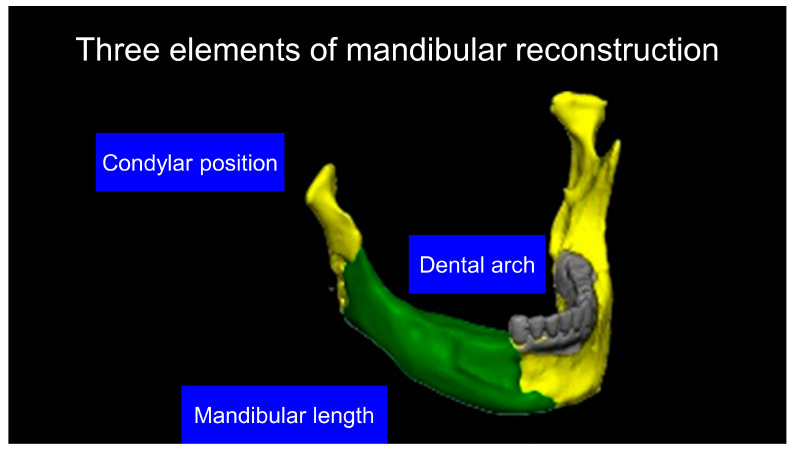
Schematic description of the mandibular reconstruction concept. The important points are condylar position, mandibular body length, and dental arch. A dental arch means that occlusion with the support of natural teeth or with dentures can be established with the optimum inter-occlusal relationship.

**Figure 4 jcm-12-01122-f004:**
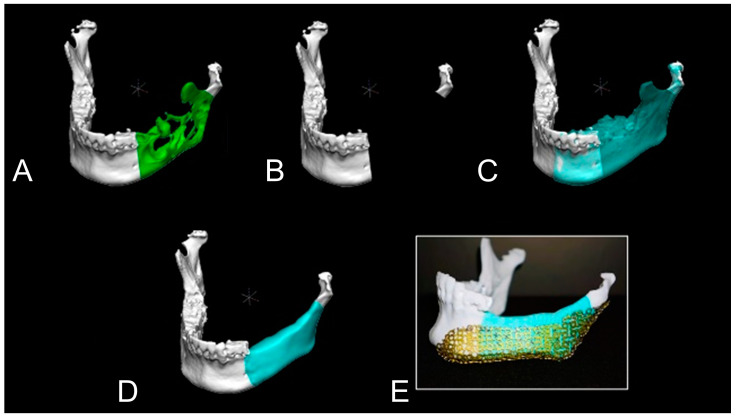
Simple mirroring technique. This case (Case 8) had ameloblastoma from the left mandibular body to ascending ramus (**A**). From the result of the preoperative imaging evaluation, the left condyle was intact. After determining the extent of the resection (**B**), the right jaw was mirrored (**C**). Then, bone augmentation was performed digitally (**D**). After that, the 3D model was produced and titanium mesh was pre-bended (**E**).

**Figure 5 jcm-12-01122-f005:**
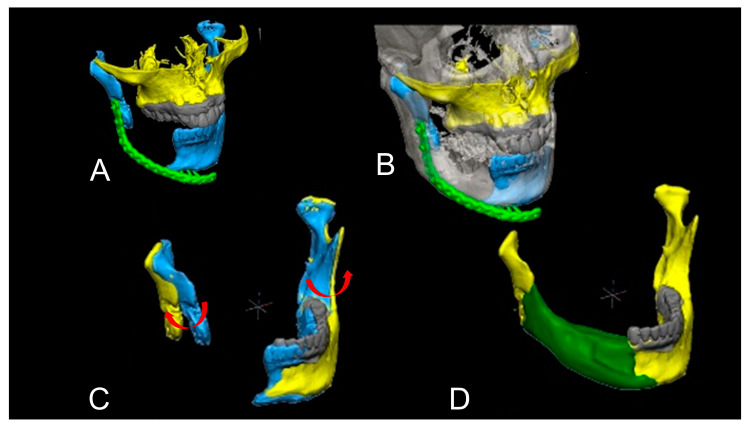
Simple mandibular deviation. This is a case of segmental mandibular defect due to ameloblastoma, and the mandible was reconstructed with a surgical plate simultaneously (Case 10). The plate was fractured, and then the right condyle and mandible were displaced (**A**). The blue mandible shows a medially deviated mandible on both sides. The yellow mandible is the ideally positioned mandible. The displaced condyle and remaining dentition were superimposed on preoperative mandible CT data. The grey mandible indicates the preoperative position of the mandibular condyle (**B**). Red arrows indicate the direction of the normal position (**C**). Thus, the exact mandibular morphology was reconstructed virtually and green indicates the bone graft (**D**). After this process, a 3D model and a positioning gauge were made.

**Figure 6 jcm-12-01122-f006:**
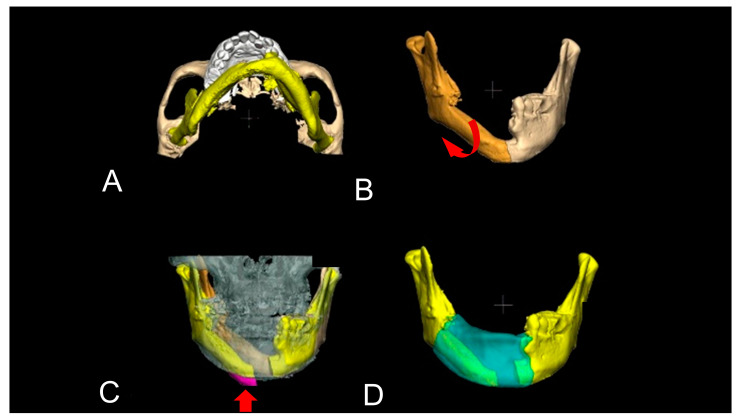
Complicated mandibular deviation with shortened and malpositioned bone graft. This is a case of segmental mandibular defect due to a tumour resection with radical neck dissection (Case 11). The mandible was reconstructed with a fibular free flap simultaneously. The yellow mandible indicates a deviated reconstructed mandible. The patient’s facial appearance was good; however, his alveolar bone for dental rehabilitation was poor (**A**). The dark brown mandible is detached at the midline and moved outwards to correct the deviation. The red arrow indicates the direction in which the detached mandible should be returned to its normal position (**B**). Preoperative CT suggested that the bilateral condyles were displaced and the length of the reconstructed mandible was unnatural (**C**). This condition made the mandible deviate to the left side and dental occlusion was poor with the remaining teeth. Thus, the first step of virtual mandibular reconstruction was to cut the reconstructed mandible (**C**). The second step was the adjustment of the displaced condylar position. Additionally, the third step was the superimposition of the remaining teeth. After these processes, the pink area of protruding bone was cut (red arrow), and the exact mandibular morphology was reconstructed virtually and green indicates the bone graft (**D**).

**Figure 7 jcm-12-01122-f007:**
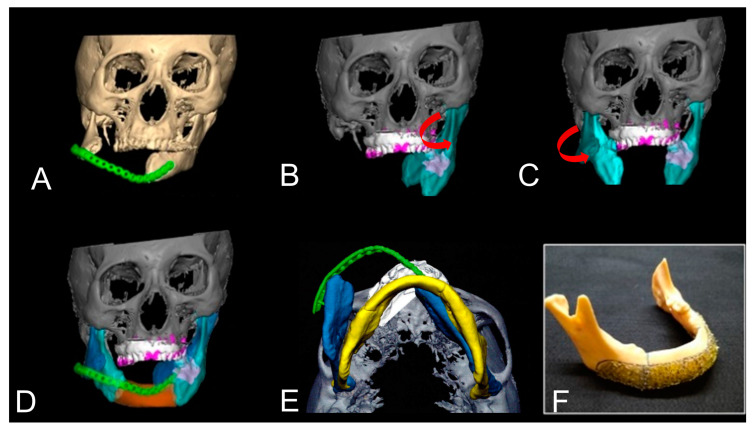
Complicated mandibular deviation without preoperative CT data. This was a case of bilateral segmental mandibular defect due to a tumour resection with radical neck dissection (Case 14). The defects were reconstructed with a surgical plate simultaneously. However, the surgical plate was located in the incorrect place and bilateral condylar displacements were seen. The facial appearance was poor and the alveolar bone for the dental rehabilitation was also improper (**A**). This condition made the remaining mandible deviate. The displaced condyles were adjusted digitally. Additionally, a dental arch of the maxilla was superimposed (**B**,**C**). The red arrow indicates the direction in which the detached mandible should be returned to its normal position (**B**,**C**). Since the information of the preoperative CT data could not be used due to the CT artefact, the virtual neo-mandible was therefore used for the superimposition (**D**,**E**). The dark brown mandible indicates a virtually grafted PCBM. The white maxilla represents the plaster model information. After these processes, the neo-mandibular morphology was reconstructed virtually. The blue mandible is the original CT data with a green surgical plate and the yellow mandible is the virtually reconstructed mandible. The positioning of the upper and lower jaws has been greatly improved (**E**). Then, a 3D model and a titanium mesh were made (**F**).

**Figure 8 jcm-12-01122-f008:**
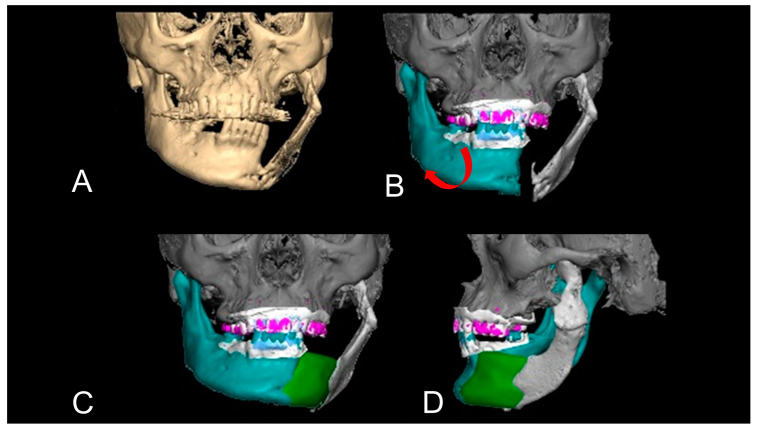
Complicated mandibular deviation with the application of the dental cast model. This was a case of a left segmental mandibular defect due to a tumour resection with radical neck dissection. The defects were reconstructed with a scapular flap and a surgical plate simultaneously (Case 15). However, the grafted bone showed non-union, and the condylar displacement was also seen. The patient’s facial appearance was poor and his dental occlusion was also impossible to bite due to the mandibular deviation (**A**). Preoperative CT suggested that the left condyle was displaced and the connection between the original mandible and grafted bone showed non-union. Moreover, the length of the reconstructed mandible was unnatural. This condition made the mandible deviate to the left side and dental occlusion was poor with the remaining teeth. Thus, the first step of virtual mandibular reconstruction was to cut the part of non-union of the mandible and the superimposition of the right condyle. The red arrow indicates the direction in which the detached mandible should be returned to its normal position (**B**). The CT data could not be used due to the CT artefact. The second step was the superimposition of a dental cast model to the alveolar portion (**B**–**D**). The white maxilla represents the plaster model information. The green mandible indicates a virtually grafted PCBM. After these processes, the mandibular morphology was reconstructed virtually with the precise position of the right dislocated condyle and mandible. This patient refused the left condylar repositioning operation. After this process, a 3D model and a positioning gauge were made.

**Figure 9 jcm-12-01122-f009:**
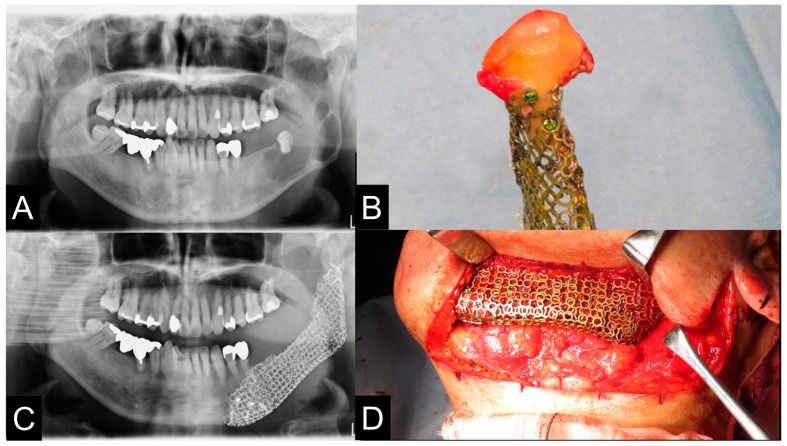
Pre- and postoperative panoramic radiographs showed accurate jaw reconstruction. (**A**,**C**). The mandibular condyle was removed once (**B**) and a PCBM was implanted directly fixed to the pre-bended TiMesh (**D**). The patient refused prosthodontic treatment because he did not feel any difficulty in masticatory function.

**Figure 10 jcm-12-01122-f010:**
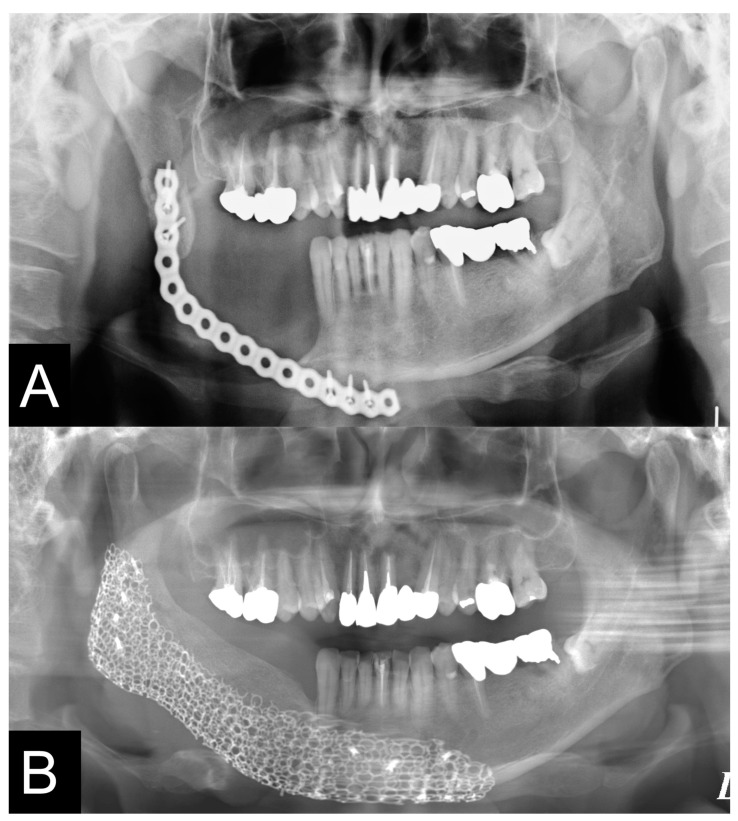
Pre- and postoperative panoramic radiographs showed accurate jaw reconstruction. (**A**,**B**). The fractured plate was removed, the pre-bended custom-made TiMesh was fixed with a positioning device, and a PCBM was transplanted. After the operation, the mandibular deviation was corrected and the position of the condyle was improved.

**Figure 11 jcm-12-01122-f011:**
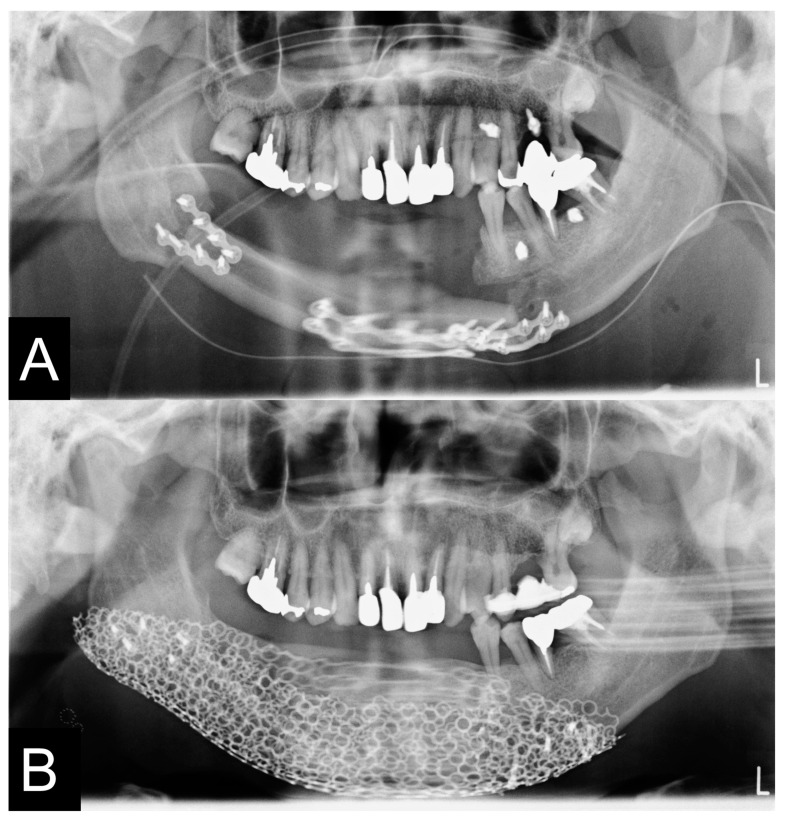
Pre- and postoperative panoramic radiographs showed accurate jaw reconstruction. (**A**,**B**). The deviated mandible was cut at the junction of the fibula and mandible with a cutting device. The patient’s pre-bended custom-made TiMesh was fixed with a positioning device and a PCBM was grafted. After the operation, the optimum alveolar crest was reconstructed for the dental implant placement and the positions of condyles were improved.

**Figure 12 jcm-12-01122-f012:**
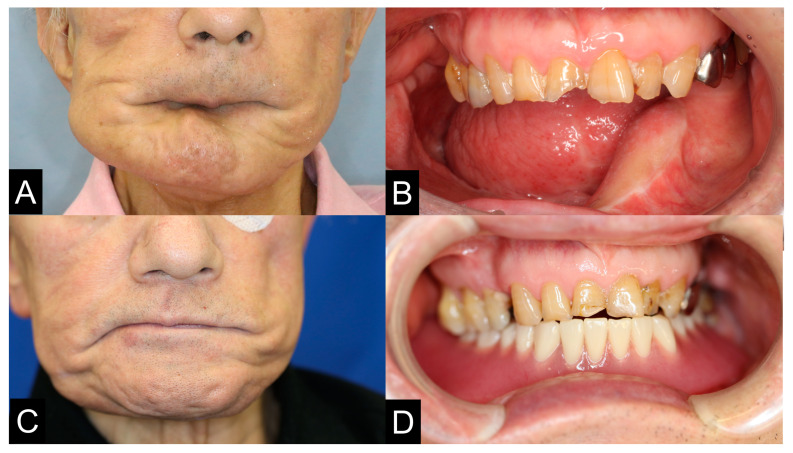
The surgical plate was removed and his pre-bended custom-made TiMesh was fixed with a positioning device and a PCBM was grafted. After the operation, the mandibular deviation has corrected and the position of the condyle was improved and the facial contour was significantly improved (**A**,**C**). Furthermore, the conventional denture could then be fitted in the mandible (**B**,**D**).

**Figure 13 jcm-12-01122-f013:**
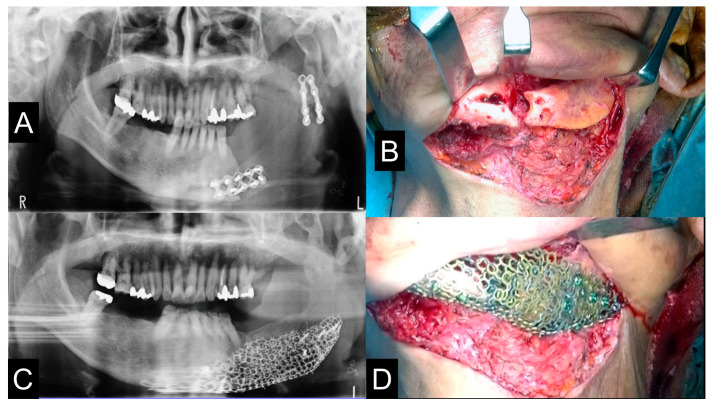
Pre- and postoperative panoramic radiographs showed accurate jaw reconstruction. (**A**,**C**). The patient’s pre-bended custom-made TiMesh was fixed with a positioning device and a PCBM was grafted (**B**,**D**). After the operation, the mandibular deviation was corrected and the dental occlusion was improved. A removal conventional partial denture was applied for occlusal rehabilitation. The left dislocated condyle did not show any clinical symptoms with ipsilateral deviation.

**Figure 14 jcm-12-01122-f014:**
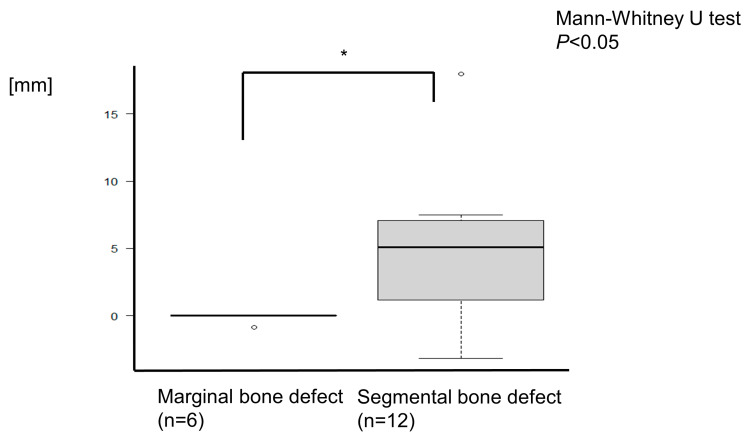
For the marginal bone augmentation, the mean length was −0.15 ± 0.37 mm, whereas for the segmental defects, the mean length was 4.89 ± 5.34 mm. The mandibular length of segmental resection was augmented longer than marginal bone resection. There was a statistically significant difference (Mann–Whitney U test, * *P* < 0.05).

**Figure 15 jcm-12-01122-f015:**
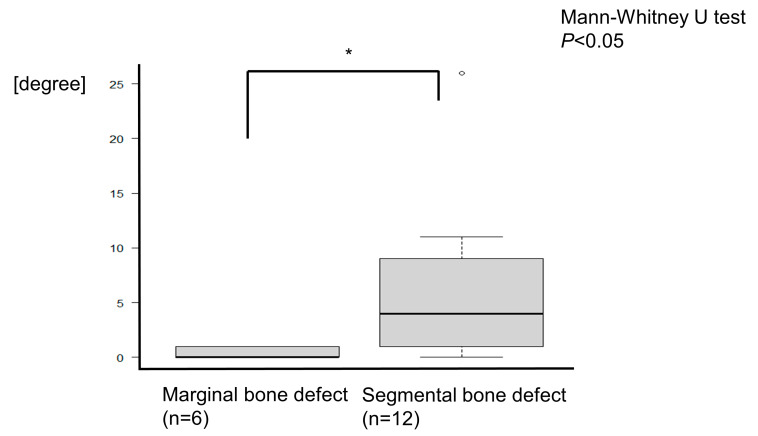
The deviation of the condyle was evaluated at the point of the condylar axial angle. For the marginal bone augmentation, the angle differences between pre- and postoperation were mean 0.33 ± 0.52°, whereas, for the segmental defects, the mean angle was 6.41 ± 7.46°. The condyle was more deviated in segmental defects than in marginal bone resection. There were statistically significant differences (Mann–Whitney U test, * *P* < 0.05).

**Figure 16 jcm-12-01122-f016:**
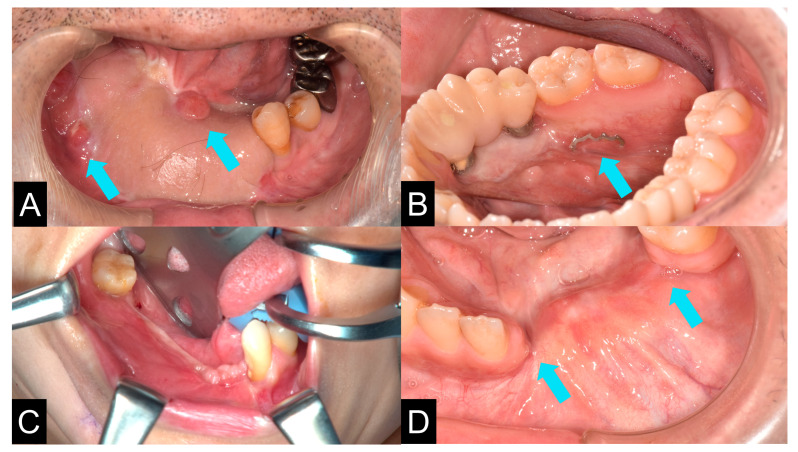
The infectious situation of the mandible (**A**). Chronic pus discharge was observed after surgery. The arrows indicate the location of the fistula and the growth of inflammatory granulation tissue. The patient underwent removal of the infected grafted bone followed by re-grafting of PCBM. Titanium mesh exposure (**B**). Arrows indicate exposed titanium mesh. The countermeasure was the removal of the exposed mesh. The narrowing of the oral vestibule after bone graft (**C**). The gap between gingival mucosa and augmented bone. Arrows indicate the gap between the alveolar bone and the grafted bone (**D**).

**Figure 17 jcm-12-01122-f017:**
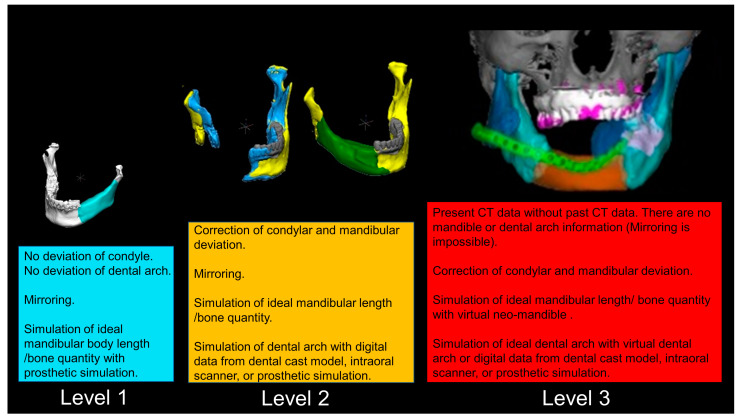
Schematic algorithm for the virtual reconstruction of the mandible with digital data. Level 1 is a relatively simple procedure. Level 2 includes deviation of the condyle and mandibular body. Level 3 is a complicated case that has lost information on original mandibular morphology with condylar and mandibular deviation.

**Table 1 jcm-12-01122-t001:** Primary patient characteristics of lesions.

Case	Gender	Age	Diagnosis of Primary Disease	Primary Reconstruction	Bone Defect	Classification Of Js Brown	Schematic of Mandibular Defect
1	M	23	Ameloblastic carcinoma	None	Marginal bone defects	Marginal bone defect	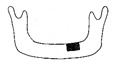
2	M	73	SCC Gingiva	Surgical plate	Segmental bone defects	Body-chin Class II	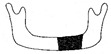
3	F	78	SCC Floor of mouth	Rectus abdominis musculocutaneous flap and surgical plate	Segmental bone defects	Body-chin Class III	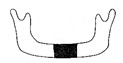
4	M	70	SCC Floor of mouth	Anterior cervical flap	Marginal bone defects	Marginal bone defect	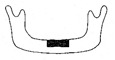
5	F	69	SCC Gingiva	Surgical plate	Segmental bone defects	Ramus-body Class I	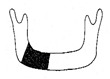
6	M	69	Osteoblastoma of the mandible	None	Marginal bone defects	Marginal bone defect	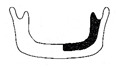
7	M	73	SCC Gingiva	None	Marginal bone defects	Marginal bone defect	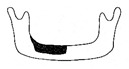
8	M	69	Ameloblastoma	Simultaneous surgery	Segmental bone defects	Ramus-body Class I	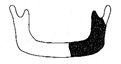
9	M	58	Mucoepidermoid carcinoma Floor of mouth	Vascularized fibular bone graft	Segmental bone defects	Ramus-body-chin Class III	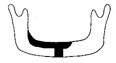
10	M	60	Ameloblastoma	Fractured surgical plate	Segmental bone defects	Ramus-body Class I	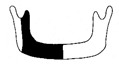
11	F	62	SCC Gingiva	Fractured surgical plate	Segmental bone defects	Ramus-body Class II	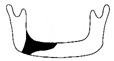
12	M	81	SCC Gingiva	None	Marginal bone defects	Marginal bone defect	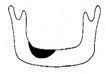
13	F	50	Odontogenic myxoma	Surgical plate	Segmental bone defects	Body-chin Class II	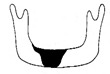
14	M	74	SCC Gingiva	Surgical plate	Segmental bone defects	Body-chin Class III	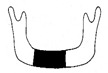
15	M	62	SCC Gingiva	Vascularized scapular bone graft (non-union)	Segmental bone defects	Body-chin Class I	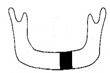
16	F	45	Ameloblastoma	Surgical plate	Segmental bone defects	Ramus-body Class I	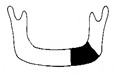
17	M	17	Ossifying fibroma	None	Mandibular partial resection	Marginal bone defect	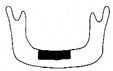
18	M	63	Ameloblastoma	Fractured surgical plate	Segmental bone defects	Ramus-body-chin Class III	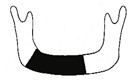

**Table 2 jcm-12-01122-t002:** The amount of PCBM simulation and realistic amount of PCBM, simulation techniques, difference of the mandibular length, and the condylar axial angle between pre- and postoperation.

Case	Virtual Reconstruction Technique	Preoperative Simulation of PCBM Amount	The Reality of PCBM Quantity	Preoperative Length (mm)	Postoperative Length (mm)	A Preoperative Sum of the Condylar Axial Angles (Degree)	A Postoperative Sum of the Condylar Axial Angles (Degree)
1	Level 1	2.2 cc	6.4 cc	26.7 mm	25.8 mm	126°	127°
2	Level 2	11.3 cc	20 cc	45.4 mm	50.4 mm	141°	151°
3	Level 2	11.9 cc	23 cc	36.0 mm	36.0 mm	150°	151°
4	Level 1	6.51 cc	10 cc	37.7 mm	37.7 mm	148°	147°
5	Level 2	11.3 cc	17 cc	46.0 mm	53.5 mm	153°	142°
6	Level 1	14.2 cc	30 cc	63.3 mm	63.3 mm	143°	143°
7	Level 1	8.2 cc	10 cc	54.6 mm	54.6 mm	131°	131°
8	Level 1	16.8 cc	17 cc	69.0 mm	69.0 mm	129°	128°
9	Level 2	20 cc	40 cc	63.4 mm	65.7 mm	106°	132°
10	Level 2	19.3 cc	40 cc	70.7 mm	77.7 mm	129°	136°
11	Level 2	10 cc	15 cc	56.3 mm	53.1 mm	136°	140°
12	Level 1	8 cc	14 cc	40.6 mm	40.6 mm	149°	149°
13	Level 2	10 cc	19 cc	35.3 mm	42.4 mm	150°	145°
14	Level 3	10.4 cc	20 cc	56.3 mm	59.6 mm	172°	161°
15	Level 3	10.6 cc	24 cc	54.5 mm	61.0 mm	197°	197°
16	Level 2	9.6 cc	21 cc	30.8 mm	36.0 mm	164°	160°
17	Level 1	5.1 cc	10 cc	44.5 mm	44.5 mm	122°	122°
18	Level 2	17.7 cc	32 cc	54.5 mm	72.5 mm	137°	140°

## Data Availability

Not applicable.

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
