# Peer review of "Towards Optimum Mandibular Reconstruction for Dental Occlusal Rehabilitation: From Preoperative Virtual Surgery to Autogenous Particulate Cancellous Bone and Marrow Graft with Custom-Made Titanium Mesh—A Retrospective Study"

_jcm, 2023, doi:10.3390/jcm12031122_

Round 1

Reviewer 1 Report

The method of virtual planning and visualization of treatment effects, especially in non-standard cases, is a valuable help for the surgeon and nowadays it is becoming more and more common, so it should also be disseminated as much as possible, because it gives a wide range of options for optimal implant adjustment, incision training or visualization of treatment effects .

The cases presented in the work are well described, but, as the Authors rightly noted in the Discussion, it would be optimal to design and manufacture using, for example, the 3D printing custom-designed bone plates and Ti mesh. 

From the medical point of view, the work is interesting and shows the positive effects of using virtual planning. However, from the engineering point of view, this method could be greatly improved by using personalized implants and surgical templates.

For this reason, I assess the novelty and significance for science as average. Nevertheless, what is missing in this work, may be described in the next article, so I accept the present work for publication.

Author Response

Reviewer #1:     The method of virtual planning and visualization of treatment effects, especially in non-standard cases, is a valuable help for the surgeon and nowadays it is becoming more and more common, so it should also be disseminated as much as possible because it gives a wide range of options for optimal implant adjustment, incision training or visualization of treatment effects.

The cases presented in the work are well described, but, as the Authors rightly noted in the Discussion, it would be optimal to design and manufacture using, for example, the 3D printing custom-designed bone plates and TiMesh.

From the medical point of view, the work is interesting and shows the positive effects of using virtual planning. However, from the engineering point of view, this method could be greatly improved by using personalized implants and surgical templates.

For this reason, I assess the novelty and significance for science as average. Nevertheless, what is missing in this work, may be described in the next article, so I accept the present work for publication.

Author:               Thank you very much for your suggestion. The manuscript on the use of digital technology to simulate alveolar bone reconstruction in complex mandibular morphology. We are currently continuing to reconstruct the occlusion with prosthetic treatment on the PCBM TiMesh reconstructed alveolar bone. Digital technology is also being used for this treatment and we are accumulating useful data. We have added an explanatory sentence at the end of the Discussion (lines 494-497).

“This study was an attempt to perform ideal alveolar bone reconstruction for better occlusal reconstruction. We are currently continuing to reconstruct the occlusal rehabilitation with prosthetic treatment by implants on the PCBM-TiMesh reconstructed alveolar bone using digital technology.”

Thank you again for your suggestions.

Reviewer 2 Report

This is an interesting topic for mandibular reconstruction but the PCBA combined with TiMesh cannot be suitable for reconstruction of extensive mandibulectomy. This modality of mandibular reconstruction limits the clinical applications and this result cannot be able to provide significant clinical significance. The title as "dental occlusal rehabilitation" was not clearly stated in manuscript and did not be considered in the study design. This is an original article but the statements is too messy as a case series so the writing skill should be improved. The description of the research results is too short to clearly state the main findings and purpose of the research. This is a clinical research so the discussion should more focus on the clinical significance and applications.

Author Response

Reviewer #2:     This is an interesting topic for mandibular reconstruction but the PCBA combined with TiMesh cannot be suitable for reconstruction of extensive mandibulectomy. This modality of mandibular reconstruction limits the clinical applications and this result cannot be able to provide significant clinical significance. The title as "dental occlusal rehabilitation" was not clearly stated in manuscript and did not be considered in the study design. This is an original article but the statements is too messy as a case series so the writing skill should be improved. The description of the research results is too short to clearly state the main findings and purpose of the research. This is a clinical research so the discussion should more focus on the clinical significance and applications.

Author:               Thank you very much for your suggestion. The reviewer is certainly right when he discusses the study design. As the reviewer pointed out, the combination of PCBM and TiMesh is not indicated for all mandibular bone reconstruction. The aim of the cases in this study was to find out how to perform alveolar bone reconstruction leading to optimum prosthetic treatment in cases with less-than-ideal jaw and alveolar bone morphology after vascularized bone graft or the other previous operation. In this context, we have chosen the title “Towards optimum mandibular reconstruction for dental occlusal rehabilitation”. The next research, which is currently in progress, is a study of a case in which an implant or conventional prosthetic treatment was placed into the reconstructed alveolar bone augmented by PCBM TiMesh. Current surgical techniques have limitations for occlusal rehabilitation. This manuscript is aimed at special cases where prosthetics could not be successfully performed with previous surgical methods. Such cases are relatively rare, hence this sample size. We hope that you will appreciate the special nature of the cases. As regards the purpose of the study you mention, it is written in lines 53-59.

We also considered that if the simulation of alveolar bone formation and the reconstruction of occlusion, such as implant prosthesis, were presented in the research at the same time, the point of argument would become unclear and the author's intention would not be conveyed. In this first report, alveolar bone reconstruction was carried out using digital technology. In a subsequent second report, a clinical study on occlusal reconstruction, including implant prosthetics, will be presented. We have added an explanatory sentence at the end of the Discussion (lines 494-497). “This study was an attempt to perform ideal alveolar bone reconstruction for better occlusal reconstruction. We are currently continuing to reconstruct the occlusal rehabilitation with prosthetic treatment by implants on the PCBM-TiMesh reconstructed alveolar bone using digital technology.”

Thank you for your suggestions.

Reviewer 3 Report

This manuscript, in my opinion, strikes a balance between its subject-matter expertise and its applicability to the academic purpose. 

Additionally, results are interpreted to emphasise the paper's thesis and effectively highlight its important points. 

1 .What is the main question addressed by the research? ANSWER-Yes it did as they have discussed all the treatment modalities of mandibular reconstruction pros and cons.     2. Do you consider the topic original or relevant in the field? Does it
address a specific gap in the field? ANSWER -Yes the topic addressed in this study has been addressed and it also adds up the knowledge on this topic.
3. What does it add to the subject area compared with other published material? ANSWER-This study strikes a balance between its subject-matter expertise and its applicability to the academic purpose   4. What specific improvements should the authors consider regarding the
methodology? What further controls should be considered? ANSWER- no such specific  improvements required 
5. Are the conclusions consistent with the evidence and arguments presented
and do they address the main question posed? ANSWER - Yes it does the conclusion and the results are relevant in there means 
6. Are the references appropriate? ANSWER- yes 
7. Please include any additional comments on the tables and figures. ANSWER- No additional improvement required

Author Response

Reviewer #3:     This manuscript, in my opinion, strikes a balance between its subject-matter expertise and its applicability to the academic purpose.

Additionally, results are interpreted to emphasise the paper's thesis and effectively highlight its important points.

1 .What is the main question addressed by the research? ANSWER-Yes it did as they have discussed all the treatment modalities of mandibular reconstruction pros and cons.     2. Do you consider the topic original or relevant in the field? Does it

address a specific gap in the field? ANSWER -Yes the topic addressed in this study has been addressed and it also adds up the knowledge on this topic.

  1. What does it add to the subject area compared with other published material? ANSWER-This study strikes a balance between its subject-matter expertise and its applicability to the academic purpose
  2. What specific improvements should the authors consider regarding the

methodology? What further controls should be considered? ANSWER- no such specific  improvements required

  1. Are the conclusions consistent with the evidence and arguments presented

and do they address the main question posed? ANSWER - Yes it does the conclusion and the results are relevant in there means

  1. Are the references appropriate? ANSWER- yes
  2. Please include any additional comments on the tables and figures. ANSWER- No additional improvement required

Author:               Thank you very much for your very encouraging comments regarding the peer review. We will continue to prepare a research article with implant treatment on this reconstructed jawbone. Thank you very much.

Round 2

Reviewer 2 Report

This revised manuscript still does not focus on the dental occlusal rehabilitation for mandibular reconstruction. I think the innovation of this study is the dental occlusal consideration in mandibular reconstruction. However, the manuscript is not clear to state how to transmit or evaluate the dental occlusion in study method. 

Author Response

Reviewer #2:    

This revised manuscript still does not focus on the dental occlusal rehabilitation for mandibular reconstruction. I think the innovation of this study is the dental occlusal consideration in mandibular reconstruction. However, the manuscript is not clear to state how to transmit or evaluate the dental occlusion in study method.

Author:              

Thank you for your suggestions. I am now aware of the seriousness of your suggestion.

In the present cases, at the time of preoperative simulation of virtual bone augmentation, simulations of occlusal reconstruction including dental implant placement were also performed simultaneously. A schematic example is shown in Figure 1c. However, not all prosthetic simulations are performed under identical conditions. Some use implant simulation software, some use a 3D printer to produce a model from digital data, and some use a cast model to confirm the occlusal positional relationship. We would like to present the details of this in a future paper. In terms of prosthetic treatment, not all implant prostheses are available due to patient financial considerations, such as whether the prosthesis is covered by our health insurance systems. Therefore, once alveolar bone augmentation has been simulated, there are limitations in simulating prosthetics in different cases.

Taking these circumstances into account, we have added the following on lines 115, 129-140.

“Mandibular data acquisition and the making of virtual neo-mandible for optimum dental occlusion”

“As well as, virtual simulations of occlusal rehabilitation were performed to establish occlusion on the computer. After virtual bone augmentation, dental implant placement simulation or virtual conventional prosthetic treatments were continued with consideration of the opposing dentition. In some cases, after the computer simulation, these data were used to confirm the occlusal relationship on a 3D printed model or a conventional individual dental cast model. The cases included a variety of prosthetic conditions with conventional dentures, dental implant prostheses or no prostheses. In some cases, the patient's requirements changed as the mandibular and alveolar reconstruction progressed with increasing patient motivation for occlusal reconstruction. As the present research aimed to evaluate the PCBM graft with TiMesh using virtual reconstruction surgery, the following research will clarify the establishment of occlusal reconstruction.”

Thank you for your very important comments. We will make full use of your advice in our future research.